# The Physical and Structural Effects of 1-MCP on Four Different Apple Cultivars during Storage

**DOI:** 10.3390/foods12224050

**Published:** 2023-11-07

**Authors:** Valentina J. L. Ting, Pat Silcock, Franco Biasioli, Phil Bremer

**Affiliations:** 1Department of Food Science, University of Otago, Dunedin 9016, New Zealand; valentinating.888@gmail.com (V.J.L.T.); pat.silcock@otago.ac.nz (P.S.); 2Research and Innovation Centre, Foundation Edmund Mach, via Mach 1, 38098 San Michele all’ Adige, TN, Italy; franco.biasioli@fmach.it

**Keywords:** texture, volatile compounds, malus domestica, µ-CT scanner

## Abstract

The impact of the ethylene inhibitor, 1-methylcyclopropene (1-MCP), on four apple cultivars (Braeburn, Fuji, Jazz and Golden Delicious) over 150 days of storage at 2 °C was assessed. Proton transfer reaction quadrupole mass spectrometry (PTR-QUAD-MS) was used to monitor changes in VOC composition, while texture analysis and X-ray microcomputer tomography (µ-CT) scanning were used to study microstructural changes. The application of 1-MCP on apples reduced VOC emissions, concurrently maintaining a firmer texture compared to the untreated apples at each time point. The µ-CT scanning revealed how changes in specific morphological characteristics such as anisotropy, connectivity and porosity, size and shape, as well as the interconnectivity of intracellular spaces (IS) influenced texture even when porosity was similar. Additionally, this study showed that the porosity and connectivity of IS were associated with VOC emission and increased simultaneously. This study highlights how the morphological parameters of an apple can help explain their ripening process during long-term storage and how their microstructure can influence the release of VOCs.

## 1. Introduction

Following harvest, an apple’s firm and crunchy texture deteriorates during long-term storage [1]. Texture deterioration occurs because apples are climacteric fruit and therefore experience a peak in respiration after fruit abscission accompanied by a characteristic increase in the production of ethylene [2]. Ripening results in the apple becoming sweeter due to the hydrolysis of starch into sugars and also increases volatile organic compound (VOC) production due to the role of ethylene as a plant growth regulator [2]. Postharvest treatments have traditionally focused on maintaining apple texture and prolonging their desirable sensorial characteristics [1,3]. 

Two commercial postharvest treatments commonly used to extend the shelf-life of apples are cold storage and the use of the ethylene inhibitor, 1-methylcyclopropene (1-MCP) [1,4]. Briefly, the use of cold temperatures during long-term storage decreases the rate of respiration in apples. As a result, normal metabolic functions which are temperature-dependent, such as ethylene production, are impeded [5]. The application of 1-MCP does not decrease ethylene production but prevents ethylene-induced responses by binding onto the ethylene receptors [6]. Although these treatments help to maintain a firm and crunchy apple texture, the production of some odour-active VOCs are reduced as the involvement of ethylene in the final stages of the VOC biosynthetic pathway is hindered [7,8].

Although the effects of cold storage and 1-MCP on the internal ethylene concentration (IEC), apple texture, acidity, anti-oxidant total polyphenol content, colour, sugar content, cell wall hydrolases, and VOC composition have been well documented [1,4,9,10,11,12,13], their effects on the microstructure of apple parenchyma is less well known. Previous research on the impact of apple microstructure has generally focused on how the morphological size and orientation of the intercellular spaces (ISs) influences texture [14,15].

An important measurement regularly associated with a soft and mealy apple texture is porosity [16]. Porosity is a percentage of the total IS within a volume and as such it does not reflect the morphology, orientation or size of the ISs, parameters that have also been shown to affect the texture of specific cultivars. Small and large interconnected ISs have been speculated to occur during storage through the formation of schizogenous and lysigenous IS, respectively [17]. However, in undamaged apples, it seems more likely that the formation of large ISs in apples during long postharvest storage is caused by the degradation of the middle lamella through enzymatic degradation, which decreases cell-to-cell adhesion, resulting in an increased IS size [16]. Lysigenous ISs are more commonly an indicator of cell degradation through cell lysis, as observed in damaged apples, such as in Braeburn browning disorder (BBD). For stored apples, 1-MCP is speculated to delay the solubilization of polyuronides and neutral sugars [18] and reduce the activity of enzymes (polygalacturonase, pectin methylesterase, cellulase, β-galactosidase, and α-L-arabinofuranosidase) that cause cell wall degradation [19].

Previously, Ting and others [17] have used an X-ray microcomputer tomographic (µ-CT) scanner to determine the morphological properties of four apple cultivars (Braeburn, Jazz, Golden Delicious and Fuji). The current study expands upon previous research by investigating how apple VOC composition, micro-structure and texture change during cold storage in the presence or absence of 1-MCP.

## 2. Materials and Methods

### 2.1. Apples

Four apple cultivars (Braeburn, Fuji, Golden Delicious and Jazz) harvested at commercial maturity based on starch index, firmness and soluble solids content (SSC) were sourced from a commercial orchard in Central Otago, New Zealand, at harvest, and stored at 2 °C under regular atmospheric conditions in a conventional cold store. All apples were graded Class 1 as a classification of good quality with shape, size and colouring characteristic to the specific cultivar [20] of export quality and sourced from the first pick. Two 20 kg boxes of apples were obtained for each cultivar. One box of apples acted as a control and the other box was commercially treated by the growers with the ethylene inhibitor, 1-MCP (will be referred to as Smart Fresh™ (SF) from this point on), within 24 h of harvest before being placed in cold storage. For all cultivars and treatments, only apples within the weight range of 165–185 g were selected for further trials. SF treated and untreated apples were stored in separate cardboard boxes lined with paper pulp fruit trays. To minimize differences within an apple cultivar [21], only apples of similar shape (within the specified weight range) and colour were visually selected. Over the course of the trial, three apples of similar colour (selecting apples with similar blush intensities for bi-coloured cultivars) from each cultivar and treatment combination were selected at days 50, 70, 100, 120, or 150 of storage and their VOC release, texture and physicochemical properties assessed. The study started at day 50 to replicate the likely time when consumers would first be able to purchase apples that have been treated with 1-MCP.

In addition, on days 50, 100, and 150, apple porosity and structure was assessed using a µ-CT scanner. Apples were brought up to room temperature 24 h before analysis. Regions of the apple sampled for the different analysis are shown in Figure 1. Preliminary results from Ting and others [17] demonstrated that triplicate measurements were sufficient to observe differences between cultivars.

### 2.2. Chemicals

Antioxidant dip solution for fresh cut samples was prepared using ascorbic acid (0.2% *w*/*w*) (Hawkins Watts, New Zealand), calcium chloride dihydrate (0.2% *w*/*w*) (Unilab, Auckland, New Zealand), citric acid anhydrous (0.5% *w*/*w*) (Absolute Ingredients Ltd., Penrose, New Zealand) and made up to 100% *w*/*w* with distilled water.

**Figure 1 foods-12-04050-f001:**
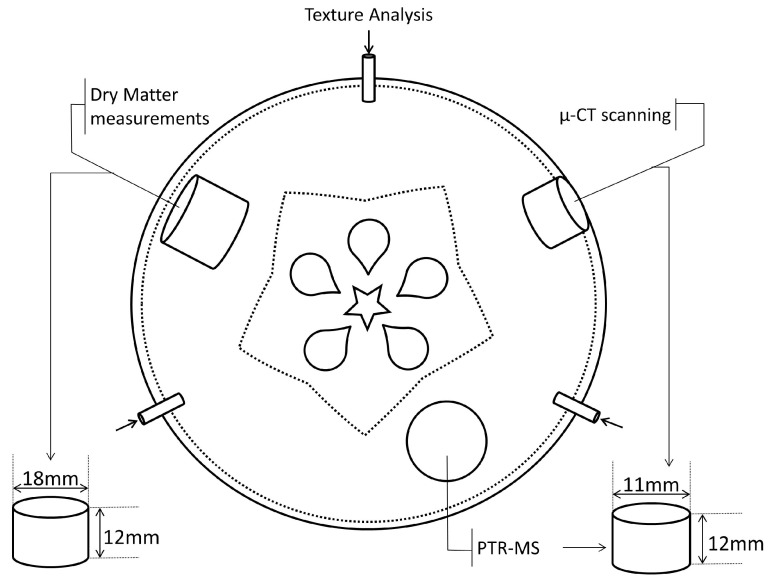
Schematic diagram of the apple regions used. A texture penetration test was carried out on three different locations along the equator of the apple. Three cylinders were cut from the apple where the larger one (without peel) was used for dry matter measurements. Two smaller cylinders of equal size were cut for µ-CT scanning (with peel) and PTR-MS measurements (without the peel parallel to the core). Three separate apples were measured for each cultivar.

### 2.3. Texture Analyzer

Apple texture was measured using a Texture Analyzer (TA-HDplus, Stable Microsystems Ltd., Goldaming, UK) fitted with a 2450 N load cell and a 6 mm flat cylinder probe attachment. Figure 1 shows the regions used for the penetration test where an entire apple was positioned in the centre of the platform and the probe was used to penetrate the equatorial section of the fruit at a speed of 1.5 mm/s for a distance of 5 mm after sensing a trigger force of 0.25 N. Texture measurements were carried out in triplicate and three apples from each cultivar were assessed. The acquired force–displacement curves were used to calculate the following parameters in Table 1, as defined by Bourne [22].

### 2.4. Soluble Solids Content and Dry Matter Concentration

The soluble solids content (SSC %) of the apples were determined by Brix using a hand refractometer (Brix 0–32%, N1, Atago, Japan) on the juice expressed from each replicate from the texture penetration test [23]. Dry matter concentration (DM %) was calculated by dividing the weight of the apple after drying by its weight before drying and expressing it as a percentage.

### 2.5. Headspace Volatile Organic Compound Measurement

Single apple cylinders (cut from the blush side of bi-coloured apples) were dipped into an antioxidant solution and placed into a 250 mL glass bottle (GL45, Duran Group GmbH, Germany) that was capped with a glass lid which was sealed using a polytetrafluoroethylene (PTFE) ring to create an air tight system. The lid contained two PTFE tubes: a gas inlet flushing instrument grade synthetic air through an active charcoal filter (Supelcarb^®^, Supelco, Bellefonte, PA, USA) at a flow rate of 50 standard cm^3^ per minute (sccm); the other connected to the proton transfer reaction quadrupole mass spectrometer (PTR-QUAD-MS) (Ionicon Analytik, GmbH, Innsbruck, Austria) inlet (~1 m, 1/16” outer Ø Silcosteel™ capillary (Restek Co, Bellefonte, PA, USA) heated to 80 °C during measurement. Apples were incubated for 30 min at 30 °C in the bottles prior to measurement.

The PTR-QUAD-MS conditions were configured in accordance with Biasioli and others [24]. A flow rate of 50 sccm over a mass ion range of *m*/*z* 20–210 at a dwell time of 100 ms (1 entire spectrum reading = 19 s) was used. The instrument was operated under the drift tube conditions of 140 Td (Td = Townsend: 10^−17^ V cm^2^ mol^−1^). Each sample was measured for 6 cycles and the mean of cycles 2–6 was used for data analysis.

### 2.6. X-ray Micro-CT Scanning and Image Analysis

Image acquisition and segmentation were carried out in accordance to Ting and others [17] using the apple cylinders represented in Figure 1. The public domain software, ImageJ version 1.49i (64-bit) [25], was used to transform the attained 16-bit cross-section images into 8-bit binary images by Otsu thresholding. A representative elemental volume (REV) of 3.0^3^ mm^3^ at a resolution of 9.89 µm/pixel was used for image analysis. IS was identified as the black foreground against the white background, which represented the apple parenchyma.

The BoneJ plugin [26] was used to measure the morphometric parameters of the IS on the processed 8-bit binary images. These are defined as: Porosity (%): Volume of calculated IS divided by total volume of selected voxel cube. Anisotropy: The degree of orientation of the substructures within a REV. The value was reported as isotropic at zero for isotropic samples. An increase in anisotropy brought the value closer to 1 (the data were multiplied by 10 to give positive log values). Connectivity: The degree to which an IS was connected assuming only one IS was present in the foreground. To do this, the plug-in ‘Purify’ (BoneJ) which acts as a filtering step was performed to locate all IS within the REV and removed all but the largest interconnected IS present. Next, the connectivity algorithm was performed, which calculated the connectivity of the image based on its calculated Euler characteristic.

### 2.7. Data Analysis

#### 2.7.1. Headspace Analysis

The PTR-MS data from the headspace analysis of the apple pieces was converted from counts per second (cps) to concentration in parts per billion per volume (ppbv). This was performed in accordance to Lindinger and others [27], assuming the proton transfer reaction rate constant to be 2 × 10^−9^ cm^3^/s for all compounds. This allows an estimation of concentration without calibration with a possible systematic error of about 20%.

Compounds that were omitted from the analysis included: *m*/*z* 21, 30, 32, 34, 37, 39, 44 which represented water clusters, saturated peaks, and machine-related compounds. Some masses such as *m*/*z* 45, 55, 57, and 63 were retained as they are isobaric compounds associated with both machine-related compounds and VOCs contributing to apple aroma. As ethylene has a proton affinity lower than H_3_O^+^, it cannot be measured directly; however, its concentration can be calculated by considering the signal at *m*/*z* 28, which derives from the reactions of ethylene with the spurious O_2_^+^ in the instrument drift tube and the ethylene concentration. Due to ethylene’s low proton affinity, it is not protonated by H_3_O^+^ but rather ionised by the charge transfer from O_2_^+^ (note that O_2_^+^ is present in low levels in H_3_O^+^). The concentration of ethylene present was therefore calculated according to Cappellin and others [28] by measuring the parent ion at *m*/*z* 28 and using a reference standard to calibrate and provide an accurate real-time estimate in ppbv.

#### 2.7.2. Statistical Analysis

A two-way analysis of variance (ANOVA) was carried out on the headspace data to evaluate the effects of storage time and treatments on different apple cultivars. One-way ANOVA was used to investigate the cultivar effects on texture, morphological, and physico-chemical properties with regard to storage duration. Variables that significantly (*p* < 0.05) differentiated cultivars based on treatment and postharvest storage duration were used for further analysis. To understand the relationship between texture, physico-chemical and morphological properties, Pearson’s correlation test at a *p* < 0.05 was also carried out. Both ANOVA and Pearson’s correlation calculations were performed using SPSS v20.0 (IBM Statistics, Inc, Chicago, IL, USA). Principal component analysis (PCA) and multi-factor analysis (MFA) was performed to investigate the interrelationships between the collected data. This comprised standardized (1/SD) PTR-MS, texture, and physico-chemical and morphological properties from day 50, 100 and 150 measurements. Both PCA and MFA were carried out using the FactoMineR package [29] in R Core version 3.2.0 [30].

## 3. Results

### 3.1. Headspace Analysis of VOCs from Fresh Cut Cylinders

#### 3.1.1. Comparing Untreated and SF-Treated Apples

A total of 38 VOCs showed a significant treatment effect (*p* < 0.05) between the untreated and SF-treated cultivars on day 50 (Table 2). VOC concentrations of each *m*/*z* for the untreated and treated apples have been reported using a tentative identification based on published literature obtained from earlier PTR-ToF-MS, PTR-MS and GC-MS measurements on apples and related products such as macerated or whole apples [31,32,33].

On day 50, the application of SF to the apples resulted in a significant reduction in the concentration of butyrate-related esters (*m*/*z* 89), isoamyl/butyrate-related esters (*m*/*z* 103), hexanoate-related esters (*m*/*z* 117), methyl hexyl (*m*/*z* 131)-related esters, butanal (*m*/*z* 73), butyl propanoate (*m*/*z* 75), ethyl hexanoate (*m*/*z* 145), alcohols (*m*/*z* 33, 47, 83, 85) and ester and alcohol fragments (*m*/*z* 41, 43, 53, 55, 57, 61) (Table 2). In contrast, SF-treated apples showed only slight reductions in their concentrations of specific aldehydes, alcohol and terpene compounds, such as trans-2-hexenal/monoterpene (*m*/*z* 81), hexanal (*m*/*z* 83), pentanal (*m*/*z* 87), farnesene fragments (*m*/*z* 95, 123, 135, 137) and α-farnesene (*m*/*z* 205). These results were in agreement with previous studies which reported that the impairment of ethylene-related pathways results in a significant reduction in the concentration of ester compounds but not the concentration of aldehydes [34,35,36].

To investigate the efficacy of SF treatments over the entire storage time, the total VOCs at each time point were plotted for each cultivar (Figure 2). At most time points, the total VOC concentration of SF apples was at least 50% lower than for untreated apples with the exception of Fuji (Figure 2). This result was comparable to results previously reported for apples [36,37] and apple juice [8]. Untreated Braeburn and Jazz cultivars showed a significant linear increase in total VOCs over time, which was predominantly driven by a large increase in acetaldehyde (*m*/*z* 45) (see Section 3.1.3). The SF-treated Braeburn cultivar showed a significant increase in total VOCs after day 100. The total VOCs in the SF-treated Jazz cultivar did not change significantly over time. The total VOC concentration for untreated Fuji apples was significantly higher than the SF-treated Fuji apples for all time points, although a significant increase in VOC concentrations overtime were not recorded. Lower total VOC concentrations for Fuji apples compared to the other three cultivars were not surprising as Fuji apples are appreciated for their ability to retain texture over extended postharvest storage rather than for flavour [38]. No significant changes in total VOCs were observed for the untreated Golden Delicious apples during storage; however, SF-treated Golden Delicious apples showed a significant decrease in total VOCs between day 50 and day 70 before starting to increase towards the levels measured on day 50 after day 70 with no significant difference between day 100 and 150. SF-treated Golden Delicious apples were lower in total VOCs compared to the untreated Golden Delicious apples at all storage time points.

#### 3.1.2. Ester Compounds

The evolution of the key fruity ester-related compounds (*m*/*z* 61, 89, 117, 145) were monitored over postharvest storage time (Figure 3), with significant changes in VOC content over time for each cultivar being reported in Table 3.

In the untreated samples, the Braeburn cultivar was the highest emitter for all ester-related VOCs (Figure 3a–d, left panels, Table 3). The Fuji cultivar was the lowest emitter for acetate ester-related VOCs (*m*/*z* 61) but not for butyrate-related esters (*m*/*z* 89), hexanoate-related esters (*m*/*z* 117), or ethyl hexanoate (*m*/*z* 145). Although Golden Delicious was the highest ethylene emitter, this trend was not observed for the ester-related compounds, possibly because Golden Delicious only produces intermediate levels of esters [39,40,41]. Higher levels of ethylene in Golden Delicious apples is reportedly due to their higher respiration rates in comparison to other cultivars such as Fuji, and it has been speculated that ethylene in Golden Delicious apples favours the regulation of cell wall-modifying enzymes related to texture softening [42]. However, more studies need to be carried out in order to understand how ethylene is utilized by Golden Delicious apples (favouring texture softening or VOC production). In comparison to the untreated samples, the SF-treated samples emitted significantly lower (*p* < 0.05) concentrations of ester-related VOCs for all cultivars across all time points (Figure 3a–d right panel, Table 3). This result was in agreement with studies on Royal Gala apples [36] and McIntosh Apple juice [8], but differed from the findings reported for Honeycrisp Apple Juice [8], a result which contributed to the Honeycrisp cultivar’s unusual response to 1-MCP treatment.

Interestingly, the suppression of ester VOC formation was not absolute, as a significant increase occurred after 120 days of cold postharvest storage for Braeburn and Golden Delicious cultivars. A faster recovery of the ester-related VOCs (*m*/*z* 89, 117, 145) for Braeburn and Fuji cultivars is due to their differences in metabolic rates and substrate availability [43]. A significant increase in ethyl hexanoate (*m*/*z* 145) was observed for the untreated Jazz cultivar but not for the treated samples.

#### 3.1.3. Aldehyde Compounds

Although SF suppressed ethylene and ester-related VOC emission (previous section), the reduction in aldehyde and alcohol VOCs was not as pronounced (Figure 4). This result was comparable to the results of Lurie and others [44], who observed a retention of alcohols and aldehyde compound emission in SF-treated Anna apples despite a decrease in ester and total VOCs. In contrast, Royal Gala cultivars treated with 1-MCP had higher hexanal and 2(E)-hexanal levels after 8 months CA storage at 0.5 °C and 7 days in air at 20 °C compared to untreated apples [36]. In the current study, for all cultivars, treatments and postharvest storage days, acetaldehyde (*m*/*z* 45) (Figure 4a) was the most abundant VOC. In the current study, only untreated Jazz and Braeburn cultivars showed a significant increase in acetaldehyde production over time (Table 3). Golden Delicious and Fuji cultivars showed no significant differences in acetaldehyde concentration throughout the storage time (Table 3). When comparing the increases in acetaldehyde (Figure 3) to the total VOCs in Figure 2, the increase in total VOC concentrations for Jazz and Braeburn cultivars was observed to be predominantly due to the increase in acetaldehyde.

Aldehyde compounds are known to be possible precursors involved in ester synthesis [7,45]. Positive correlations were observed (*p* < 0.05; mean r = 0.71) between butanal (*m*/*z* 73) and butyrate-related esters (*m*/*z* 89); and between hexanoate-related esters (*m*/*z* 117) and butyl butanoates (*m*/*z* 145), indicating their contribution to ester synthesis. However, hexanal/2-hexanone (*m*/*z* 101) was also positively correlated (*p* < 0.05; mean r = 0.84) to hexanoate-related esters (*m*/*z* 117) and ethyl hexanoate (*m*/*z* 145). Both compounds appear to contribute to the formation of esters.

#### 3.1.4. Alcohol Compounds

There was a significant linear increase in alcohol emission in the untreated Braeburn and Jazz cultivars (Figure 5a,b left panel) where significant changes at each time point are shown in Table 3. Untreated Fuji and Golden Delicious cultivars showed a significant increase in ethanol (*m*/*z* 47) but not methanol. Ethanol and methanol are commonly produced in apples during ripening [46,47]; however, studies on alcohol production on 1-MCP-treated apple cultivars stored specifically under regular atmospheric conditions are scarce. A study investigating the metabolic changes in 1-MCP-treated Empire apples during controlled atmosphere storage reported a decrease in ethanol production for 1-MCP-treated apples and an increase in methanol production regardless of treatment [48]. In this study, both methanol and ethanol increased over time for some untreated cultivars, but was suppressed until a certain time point in the SF-treated cultivars (Figure 5a,b right panel, Table 3) indicating that the application of SF slowed the rate of increase in ethanol and methanol concentrations. However, these compounds subsequently showed a significant cultivar-dependent increase after 70–100 days. The evolution of ethanol and methanol can be considered a spoilage indicator, as these compounds can be associated with fermentation, which generally only occurs when an apple has been exposed to anaerobic conditions during long-term storage. In the current study, the apples did not show signs of fermentation.

### 3.2. Texture, Physico-Chemical and µ-CT Results

The texture, physico-chemical and morphological data for the days that coincided with the µ-CT scanning (days 50, 100, 150) were subjected to a Pearson’s correlation (Table 4). Comparing the texture and µ-CT data, porosity was negatively correlated to Fmax (r = −0.691), flesh firmness (r = −0.715) and AUC flesh (r = −0.811), as previously reported by Ting and others [17], in which examples of images obtained from apples using µ-CT scanning can be viewed. These findings were also consistent with previous findings that apples with a higher porosity have a soft, mealy texture [16,49].

Although the textural differences of some cultivars with similar porosities can be explained by studying the IS size distribution [17], how the different-sized ISs relate to textural stability is still unclear, as is whether a large IS occurs as a result of smaller ISs connecting, and how IS morphology relates to porosity. To gain further information on IS orientation and shape, the anisotropy and connectivity of the samples were examined. Anisotropy defines the orientation of IS within a selected REV based on the mean intercept length (MIL) method, and is obtained as scalar values between 0 (isotropic) and 1 (anisotropic) [50]. To understand the concept of isotropy and anisotropy, a simple analogy can be used. Consider a cube containing different types of solid materials. If this cube contains marbles of similar size, a line that passes through the cube at any orientation would make a similar number of intercepts. This cube can then be said to be isotropic. In relation to the apple microstructure, this means the absence or presence of IS along a specific directional axis is symmetrical and could be associated with a few large IS within a selected microstructural REV representing the absence of structures along the directional axis.

However, if the same cube contained a bunch of steel rods, the lines running parallel to these rods will encounter fewer boundaries compared to lines perpendicular to the rods. This increases the anisotropic nature of the ISs in the cube. The anisotropic nature of the ISs within a microstructure increases when a larger chance of intercept may occur. This is represented by smaller ISs which are randomly distributed throughout the apple parenchyma. Therefore, the negative correlation between anisotropy and porosity (Table 4, *p* < 0.05; r = −0.619) indicated that cultivars with larger porosity were less anisotropic with ISs of larger size. The positive correlation between anisotropy and AUC flesh (*p* < 0.05, r = 0.654) suggested that apples which tended to be anisotropic were firmer, indicating a possible influence of anisotropy on the mechanical properties of apple parenchyma.

Connectivity, which is the number of connected ISs within the microstructure, was negatively correlated with anisotropy (*p* < 0.05; r = −0.745) and positively correlated with porosity (*p* < 0.05; r = 0.493). These data showed that apple parenchyma with a high connectivity was more porous and less anisotropic. Previously, the textural differences between cultivars of similar porosities were described subjectively with the use of 2D and 3D images [17]. However, in this study, a more objective approach incorporating anisotropy, connectivity, and porosity measurements were used to understand the influence of microstructure on texture, as elaborated in Section 3.2.2.

To understand the relationship between texture and microstructure on the effects of VOC release, Pearson’s correlations were carried out. It was observed that acetaldehyde (*m*/*z* 45) and ethanol (*m*/*z* 47) were correlated to both Fmax (*p* < 0.05; mean r = −0.65) and porosity (*p* < 0.05; mean r = 0.43). This could mean Fmax decreased as the concentrations of acetaldehyde and ethanol increased, as previously reported to occur during ripening (i.e., texture softening and increase in VOC). Concomitantly, during ripening, apple flesh softens and flesh porosity increases [45,47,51,52,53]. DM was positively correlated to SSC (*p* < 0.05; r = 0.789) as SSC is the product of starch hydrolysis during ripening [54].

#### 3.2.1. Relation between Porosity and Flesh Firmness

We previously reported that cultivars (Golden Delicious and Fuji apples) with a similar porosity could be significantly different in flesh firmness [17]. However, these results were limited to one time point. In the current study, the relationship between porosity and flesh firmness was assessed over long-term postharvest storage and for SF treated and untreated apples. A correlation plot on porosity and flesh firmness was used to visualize differences between cultivar and treatments. Looking at the individual cultivars (Figure 6), Golden Delicious was clustered away from the other cultivars as it was the softest and most porous cultivar. Jazz, Fuji and Braeburn cultivars were grouped in the lower right-hand corner, indicating they were firmer and less porous cultivars than Golden Delicious.

Regardless of treatment, Golden Delicious flesh firmness was negatively correlated (*p* < 0.05; r = −0.774) to porosity. Generally, an increase in porosity is associated with a decrease in firmness [17] and could be attributed to the presence of large ISs [16]. A weaker negative correlation (*p* < 0.05; r = −0.452) was observed for the untreated Golden Delicious apples compared to the SF-treated apples (r = −0.779), because SF-treated Golden Delicious apples were firmer at day 50 compared to all other Golden Delicious samples. Dramatic softening for treating Golden Delicious apples observed between day 50–100 was due to the lack of SF re-application [6]. The correlation between flesh firmness and porosity for untreated and SF-treated Braeburn apples was less obvious (r = −0.359). However, the negative correlation was stronger (r = −0.725) when data points from only the untreated samples were plotted, which indicated an increase in porosity as the flesh firmness decreased with an increased storage time. When compared to untreated Braeburn samples, no obvious correlation (r = −0.143) for SF-treated samples was observed, suggested that the application of SF maintained the apple texture. Similarly to Braeburn, SF treated and untreated Jazz cultivar samples were separated based on their porosity and flesh firmness. SF-treated samples maintained flesh firmness better than the untreated samples with a smaller r (−0.313) compared to the untreated samples (r = −0.563). The Jazz cultivar, also known as Scifresh, are cultivars that maintain a firm texture throughout long-term postharvest storage [55]. Therefore, the last time point at 150 days may not have been long enough to observe a significant change in the IS or firmness of the Jazz apples. Compared to the other cultivars, Fuji showed no correlation between porosity and flesh firmness. Although the SF-treated samples were firmer than the untreated Fuji apples, all samples were comparable in porosity. This result had previously been observed by Ting and others [17] and is due to the high porosity within the Fuji cultivar being the summation of numerous smaller ISs calculated at each time point.

#### 3.2.2. The Relationship between Connectivity and Anisotropy to Porosity

Apart from the IS size distribution, other factors may influence the mechanical properties of the parenchyma. Although the Fuji cultivar was significantly different in firmness at different time points, there was little change in porosity, indicating a change in the IS within the microstructure despite constant porosity values [56]. At day 50, Fuji apples were more anisotropic regardless of treatment and became more isotropic as they matured, indicating that the distribution of the ISs at day 50 was more sporadic within the selected REV and possibly smaller despite the apples having similar porosity values. This effect increased the probability of the occurrence of intercepts between ISs and cell walls. However, as the fruit matured, the IS volume increased, likely due to the breakdown of the middle lamella, through enzymatic degradation, which lead to a decrease in cell-to-cell contact, rounder cells, and increased IS between cells [57]. These changes shift the IS characteristics to be more isotropic as fewer intercepts will occur.

**Figure 6 foods-12-04050-f006:**
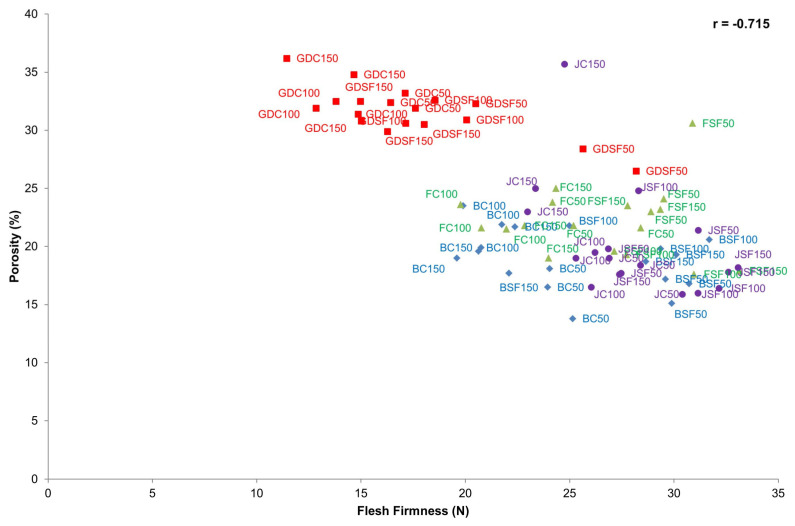
Correlation plot of flesh firmness and porosity of all cultivars, treatments, and days. Cultivars are separated based on different markers and colours, where Golden Delicious apples (GD) are denoted in red; Jazz apples (J) are denoted in purple; Fuji apples (F) are denoted in green; and Braeburn apples (B) are denoted in blue. Pearson’s correlation coefficient (r) = −0.715.

The calculation of anisotropy based on the MIL could indicate an increase in connectivity between ISs, because as the ISs become more isotropic, the MILs of all vectors have a similar meaning to a more isotropic sample, which will form a more spherical vector cloud as the anisotropy is calculated. In comparison, samples of higher anisotropy will tend to form an ellipsoid vector cloud indicative of the different vector lengths associated with a higher number of cell–ISs intercepts [26,50]. The correlation between anisotropy and connectivity for the Fuji cultivar was negative (r = −0.76), indicating that the ISs of anisotropic samples were less well connected.

### 3.3. Understanding the Inter-Relationships between VOC Release, Texture, Physico-Chemical, and Morphological Properties

To understand the interrelationships between VOC release, texture, physico-chemical, and morphological properties, a PCA was carried out (Figure 7). A total variance of 66.68% between PC1 and 2 was obtained. PC1 separated the samples based on VOCs and texture properties. Most of the samples that were highly associated with VOCs were untreated (C) samples, with the SF-treated samples being well associated with textural attributes such as the AUC flesh, flesh firmness and Fmax of the Braeburn and Jazz cultivars. Positive PC1 was dominated by untreated (C) samples with the exception of Jazz, Braeburn and Fuji cultivars at Day 50. SF-treated cultivars were mainly associated with negative PC1; however, after a prolonged storage (day 100–150), it appeared that SF-treated samples moved towards being positively associated with PC1. This was observed for SF-treated Braeburn and Fuji cultivars.

Anisotropy was positively associated with the textural properties, Fmax, flesh firmness and AUC flesh (mean r = 0.52), and negatively associated with connectivity and porosity (mean r = −0.68), indicating that cultivars which were measured as anisotropic contained the ISs of irregular sizes and orientation. Porosity and connectivity were well associated with Golden Delicious cultivars, particularly with the untreated samples. Connectivity and porosity were negatively associated with anisotropy and AUC flesh, indicating that cultivars that were porous, containing interconnected ISs, which, in turn, decrease the firmness of the cultivar. Although connectivity and porosity had a small loading on positive PC1, which was associated with VOC abundance, whether or not these properties influenced the release of VOCs is still unclear. This is because of the diversity of the cultivars selected for the experiment. For example, Braeburn and Golden Delicious cultivars were similar in selected VOC concentrations; however, Braeburn cultivars were not strongly associated with connectivity and porosity due to their firm texture. Hence, both connectivity and porosity may influence texture and VOC release. However, the latter is also strongly dependent on substrate availability that is specific to each cultivar for VOC formation [58].

Focusing on PC2, cultivars were separated into two groups; Jazz–Braeburn and Golden Delicious–Fuji. Positive PC2 separated the cultivars based on VOC composition and texture. Jazz and Braeburn apples had a firm texture and were more associated with esters, ester-related fragments and alcohol (ethanol and methanol) VOCs. Negative PC2 contained Golden Delicious and Fuji apples that were more associated with terpene-related compounds, ethylene, SSC, porosity and connectivity.

### 3.4. MFA

To understand the differences between cultivars at different days based on their treatments, two separate multi-factor analyses (MFAs) were performed (Figure 8). An MFA is a quick way to visualize the behaviour of the cultivars at a given time point [59]. In Figure 8, each coloured dot represents a different time point (red = Day 50, green = Day 100, blue = Day 150), and the legend indicates untreated (C) or SF-treated (SF) cultivars (e.g., C50 indicates untreated cultivars on day 50). The black point plotted within the individual cultivar samples is the midpoint of all coloured points. This can be used to evaluate the overall similarities or differences in these cultivars at a certain time point in a glance. If the coloured dots for a cultivar are close together, this means that large changes were not observed for that cultivar during a storage time.

The MFA of the untreated samples explained a larger variance between the samples (80.1%) compared to the SF-treated samples (72.48%) (Figure 8). To understand the agreement between the measured time points, a regression vector (Rv) was used. Here, an Rv value approaching 1 indicates that the samples are more similar, whereas an Rv value approaching zero reflects larger differences between the samples at a specific time point [59]. The Rv coefficient for the untreated samples between days 50 and 100 was 0.73. The coefficient increased for days 100 and 150 (Rv = 0.89), indicating that, regardless of the cultivar, VOC emission and texture were more different between days 50 and 100 for untreated apples than they were between 100 and 150 days. Focusing on the cultivar differences, untreated Golden Delicious was separated from all the other cultivars on Dim 1, whereas Dim 2 separates untreated Golden Delicious and Fuji cultivars from Jazz and Braeburn. Golden Delicious and Fuji had larger distances between coloured dots that represented days 50 and 150, suggesting more differences for these cultivars based on time. On the other hand, a small distance between days 100 and 150 for the Jazz cultivar represents a higher degree of similarity for the variables measured.

In Figure 8, the treated Jazz and Braeburn cultivars were separated from Golden Delicious and Fuji cultivars on Dim 2. The small distance between treated Jazz and Braeburn suggests strong similarities in terms of VOC release, morphology, texture, and physico-chemical properties compared to the Golden Delicious and Fuji cultivars. The Jazz cultivar also showed little change in the aforementioned attributes, as shown by the small distances between the coloured dots that represent storage time. The Rv coefficient between SF50 and SF100 was 0.92, indicating that treated apples changed very little within that time frame. However, the data between SF100 and SF150 were less similar (Rv = 0.71), suggesting changes that occurred in VOC release and texture data during this time in at least some apple cultivars. When comparing both SF and untreated samples, both were similar on day 50 (Rv = 0.81). Based on different time points, C100 was most similar to SF150 (Rv = 0.95), suggesting that the SF treatment could lengthen the shelf life of samples by at least 6 weeks.

## 4. Conclusions

The effect of SF treatment on the VOC release, texture and physico-chemical properties of four different apple cultivars (Braeburn, Golden Delicious and Jazz apples) has been evaluated. SF-treated apple fruits produced less VOCs compared to the untreated samples at each time point. A minimal difference in total VOC concentration was found between the SF-treated and the untreated Fuji samples, suggesting a cultivar dependence on the efficacy of SF which is better suited for cultivars with a high internal ethylene content (e.g., Golden Delicious apples).

This study also highlights the relationship between apple morphology and texture. Porosity was negatively correlated to flesh firmness, indicating the influence of microstructure on texture. An inverse relationship between flesh firmness and porosity was observed for untreated Braeburn and Jazz cultivars. The relationship between apple IS morphology and texture was shown for the first time. Previously, porosity has been used as an indicator for flesh firmness; however, this study showed that apples may have a similar porosity but have differing flesh firmness, due to morphological differences in IS, which were characterized as being lower in anisotropy and higher in connectivity, suggesting that the orientation, shape, and size of the IS, as well as IS interconnectivity, can influence texture even when porosity is similar. Porosity and connectivity were associated with the emission of VOCs, which increased concomitantly, demonstrating that specific morphological parameters could be related to ripening during long-term storage and could potentially be used to understand the impact of microstructure on VOC release. Importantly, morphological changes in apples can be measured using µ-CT scanning and used to explain how anisotropy, connectivity and porosity affect texture.

Overall, the largest change in measured attributes for the untreated samples occurred from days 50 and 100 compared to the SF-treated samples, which showed the greatest change from days 100 to 150 of storage. The efficacy of SF decreases during prolonged storage, suggesting the need for reapplication to maintain the quality attributes of specific cultivars. Nevertheless, SF-treated samples at day 150 were comparable to untreated samples at day 100. This underscores the efficacy of SF treatment in retaining the apple quality for at least 6 weeks for some cultivars.

## Figures and Tables

**Figure 2 foods-12-04050-f002:**
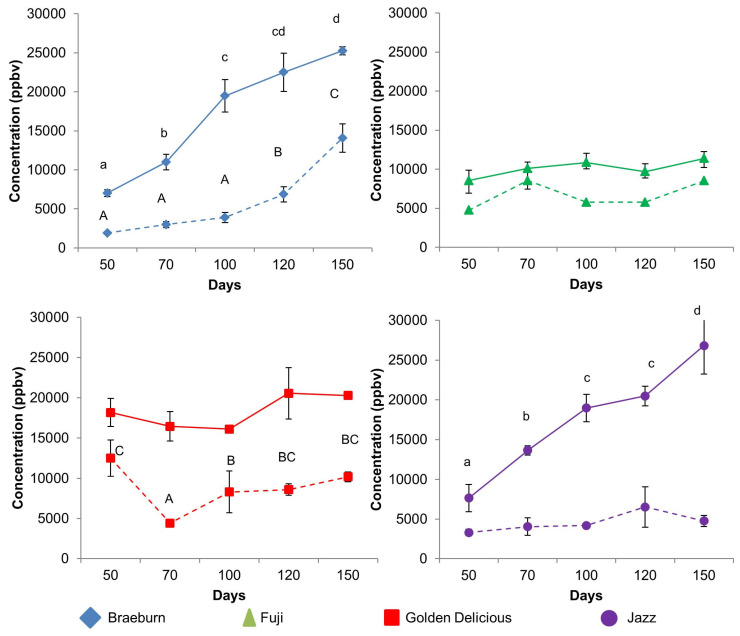
Total VOC concentration for all cultivars at 5 different postharvest days in which untreated apples are shown as continuous lines and 1-MCP treated (SF) apples are shown as dotted lines. Different letters were used to indicate only cultivars that were significantly different (*p* < 0.05) between days.

**Figure 3 foods-12-04050-f003:**
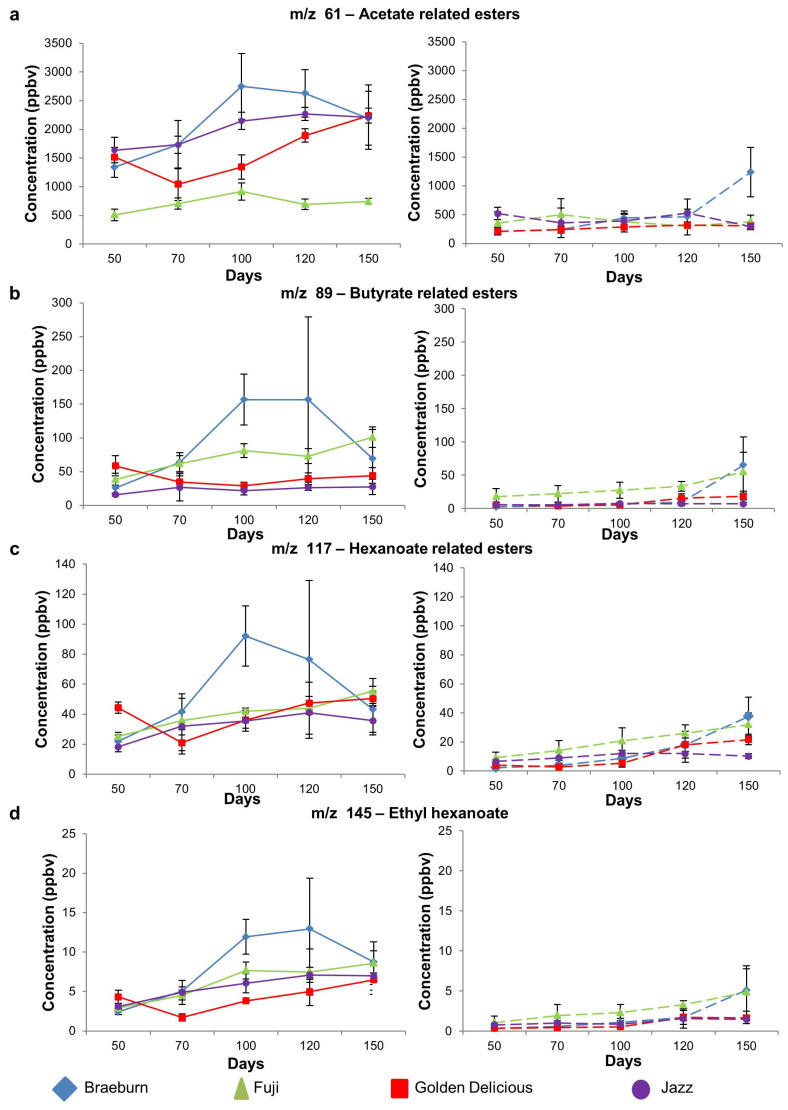
Selected ester-related mass ions (mean ± SD) for untreated (left panel) and 1-MCP treated (SF) (right panel) apple cultivars. (**a**). *m*/*z* 61–acetate related esters; (**b**). *m*/*z* 89–butyrate related esters; (**c**). *m*/*z* 117–hexanoate related esters; (**d**). *m*/*z* 145–ethyl hexanoate. Significant changes (*p* < 0.05) between days that occur for a cultivar are indicated by different letters in Table 3.

**Figure 4 foods-12-04050-f004:**
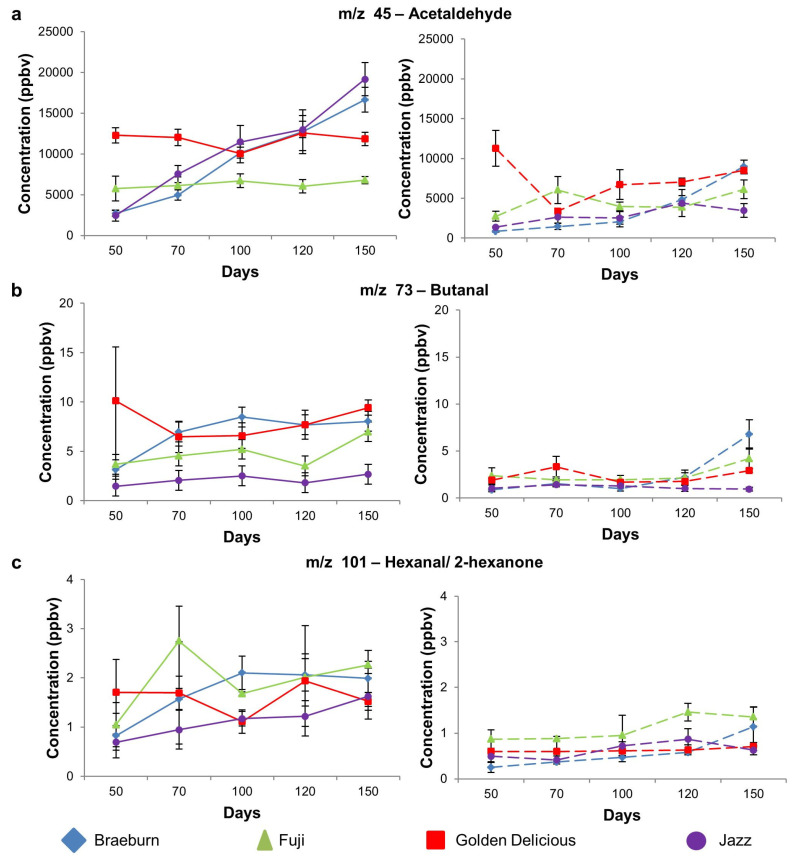
Selected aldehyde-related mass ions (mean ± SD) for untreated (left panel) and 1-MCP treated (right panel) apple cultivars. (**a**). *m*/*z* 45–acetaldehyde; (**b**). *m*/*z* 73–butanal; (**c**). *m*/*z* 101–hexanal/2-hexanone. Significant changes (*p* < 0.05) between days that occur for a cultivar are indicated by different letters in Table 3.

**Figure 5 foods-12-04050-f005:**
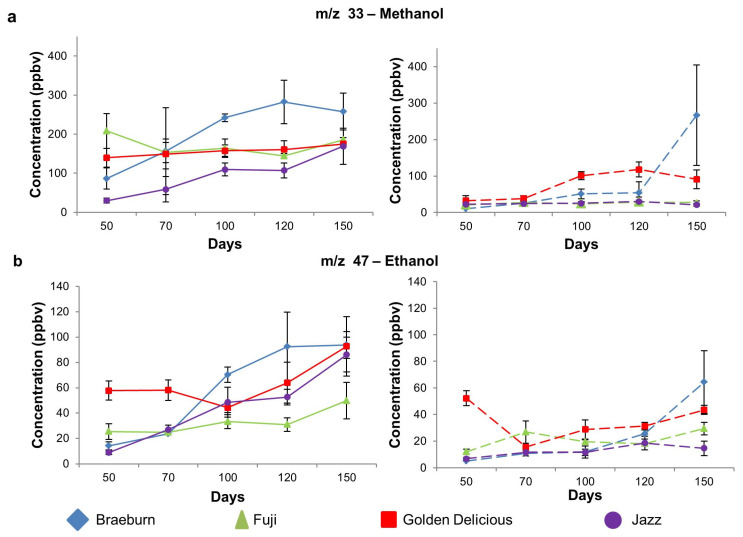
Selected alcohol related *m*/*z* (mean ± SD) for untreated (left panel) and treated (right panel) apple cultivars. (**a**). *m*/*z* 33–methanol; (**b**). *m*/*z* 47–ethanol. Significant changes (*p* < 0.05) between days that occur for a cultivar are indicated by different letters in Table 3.

**Figure 7 foods-12-04050-f007:**
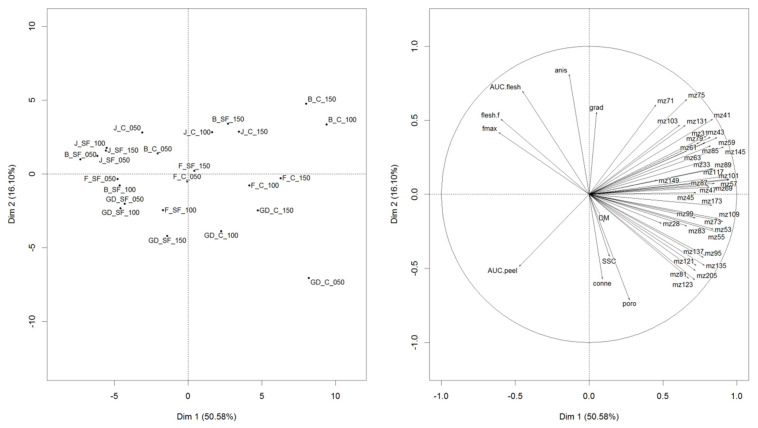
PCA of VOCs, texture, physico-chemical and µ-CT data based on the averaged triplicate data for each sample. Abbreviations: B-Braeburn; F-Fuji; GD-Golden Delicious; J-Jazz; SF-Smart Fresh; C-control/untreated.

**Figure 8 foods-12-04050-f008:**
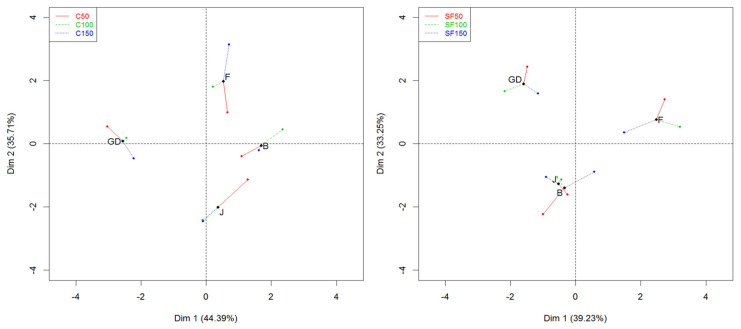
Two separate MFAs performed on all collected data to depict individual cultivar differences based on days (50, 100, 150), and compared between untreated (C) and treated (SF) samples.

**Table 1 foods-12-04050-t001:** Mechanical texture parameters and their definitions.

Mechanical Parameters	Definition
Fmax, (N)	Maximum force required to puncture the fruit skin
Flesh firmness, flesh.F (N)	Averaged force measured after skin rupture
Gradient, grad (N/mm)	Stiffness of skin measured as a slope from the start of the curve until Fmax
Area under the curve, AUC peel (N⋅mm)	Mechanical work needed to reach the rupture point of the skin indicated by Fmax taken as the area under the curve
AUC flesh (N⋅mm)	Work measured under the curve after skin rupture

**Table 2 foods-12-04050-t002:** Headspace concentrations (ppbv) and their standard deviations in brackets of the VOCs that significantly differentiated untreated (C) and 1-MCP-treated (SF) apples at day 50 using the PTR-MS. *p*-values for each VOC are listed comparing the untreated and treated cultivars. For significant *p*-values (*p* < 0.05), different superscripts indicate significant differences between cultivars. (Frag. = fragment).

*m*/*z*	Tentatively Identified Compounds	Untreated (C)					Treated (SF)				
		Braeburn	Fuji	Golden Delicious	Jazz	*p*-Value	Braeburn	Fuji	Golden Delicious	Jazz	*p*-Value
28	Ethylene	464	321	775	850	0.36	288	525	157	262	0.49
		(191)	(118)	(399)	(564)		(204)	(267)	(115)	(229)	
31	CH_3_O^+^	0.5 ^a^	1.3 ^ab^	0.8 ^ab^	1.4 ^b^	0.04	0.2	0.4	0.1	0.2	0.62
		(0.3)	(0.5)	(0.3)	(0.1)		(0.3)	(0.5)	(0.2)	(0.1)	
33	Methanol	87 ^ab^	208 ^c^	140 ^bc^	30 ^a^	0.00	11 ^a^	21 ^ab^	33 ^b^	23 ^ab^	0.03
		(27)	(44)	(24)	(6.1)		(0.1)	(0.4)	(13.1)	(3.7)	
41	Alcohol and ester frag.	343	358	353	393	0.67	119	248	138	214	0.04
		(28)	(91)	(47)	(30)		(28)	(81)	(20)	(49)	
43	Alcohol and ester frag.	1526 ^b^	755 ^a^	1938 ^b^	1840 ^b^	0.00	286	508	373	639	0.86
		(192)	(141)	(199)	(264)		(80)	(248)	(54)	(143)	
45	Acetaldehyde	2730 ^a^	5755 ^b^	12298 ^b^	2468 ^a^	0.00	837 ^a^	2732 ^a^	11271 ^b^	1388 ^a^	0.00
		(365)	(1524)	(947)	(688)		(82)	(621)	(2246)	(41)	
47	Ethanol	14 ^ab^	25 ^b^	58 ^c^	9.0 ^a^	0.00	5.1 ^a^	12 ^a^	52 ^b^	6.8 ^a^	0.00
		(3.2)	(6.1)	(7.5)	(1.5)		(0.2)	(1.8)	(5.6)	(0.9)	
53	C_4_H_5_^+^	2.6 ^a^	3.8 ^ab^	5.4 ^b^	2.0 ^a^	0.13	1.0 ^a^	2.0 ^b^	1.2 ^a^	1.1 ^a^	0.01
		(0.5)	(0.2)	(1.9)	(0.3)		(0.1)	(0.2)	(0.5)	(0.2)	
55	C_4_H_7_^+^	69 ^a^	88 ^ab^	148 ^b^	42 ^a^	0.10	25 ^a^	43 ^b^	38 ^ab^	28 ^ab^	0.03
		(8.2)	(10)	(55)	(1.9)		(1.2)	(5.4)	(8.1)	(8.3)	
57	Alcohol and ester frag.	207 ^a^	221 ^a^	487 ^b^	219 ^a^	0.00	47 ^a^	150 ^b^	116 ^ab^	101 ^ab^	0.03
		(28)	(35)	(77)	(13)		(11)	(53)	(30)	(27)	
59	Acetone	23 ^a^	29 ^ab^	32 ^b^	28 ^ab^	0.67	12	14	19	21	0.07
		(2)	(3.9)	(4.7)	(2.8)		(0.8)	(2.2)	(1.6)	(6.9)	
61	Frag. of acetate esters/acetic acid	1339 ^b^	509 ^a^	1518 ^b^	1638 ^b^	0.00	211	350	211	519	0.04
		(176)	(102)	(165)	(221)		(65)	(199)	(42)	(108)	
63	Ethylene glycol	7.7 ^a^	6.6 ^a^	25 ^b^	9.0 ^a^	0.00	1.9 ^a^	4.0 ^a^	19 ^b^	4.1 ^a^	0.00
		(0.8)	(1.1)	(2.1)	(1.1)		(0.3)	(2.1)	(3.4)	(0.5)	
69	Isoprene	4.1 ^a^	4.3 ^a^	8.2 ^b^	2.8 ^a^	0.00	1.8 ^a^	1.8 ^a^	4.1 ^b^	2.3 ^a^	0.00
		(0.7)	(0.8)	(1.9)	(0.3)		(0.5)	(0.2)	(0.5)	(0.4)	
71	Alcohol and ester frag.	53 ^b^	55 ^b^	24 ^a^	42 ^ab^	0.00	27	41	25	36	0.15
		(3.9)	(14)	(4.1)	(0.9)		(7.3)	(11)	(1.7)	(10)	
73	Butanal	3.2 ^ab^	3.7 ^ab^	10 ^b^	1.5 ^a^	0.02	0.9 ^a^	2.4 ^b^	1.9 ^ab^	1.0 ^a^	0.15
		(0.8)	(0.4)	(5.4)	(0.3)		(0.2)	(0.9)	(0.2)	(0.4)	
75	Butyl propanoate	22 ^a^	24 ^a^	14 ^a^	49 ^b^	0.00	1.5	19	2	11	0.09
		(6.6)	(2.2)	(1.1)	(8.7)		(0.9)	(16)	(1.2)	(5.2)	
79	C_2_H_7_O_3_^+^	1.3 ^a^	0.6 ^a^	3.0 ^b^	1.5 ^a^	0.00	0.3	0.4	0.4	0.6	0.56
		(0.6)	(0.3)	(0.1)	(0.5)		(0.1)	(0.3)	(0.2)	(0.4)	
81	Terpene-related frag., aldehydes (trans-2-hexenal)	9.8 ^a^	15 ^a^	44 ^b^	4.9 ^a^	0.00	5.5 ^ab^	11 ^b^	10 ^ab^	3.5 ^a^	0.02
		(0.8)	(2)	(11)	(1.6)		(0.9)	(2)	(4.4)	(1.9)	
83	Alcohols (hexanal, trans-2-hexenol, cis-2-hexenol)	18 ^ab^	26 ^ab^	31 ^b^	13 ^a^	0.02	8.2	15	12	10	0.18
		(2.9)	(2.9)	(11)	(0.8)		(0.3)	(3)	(4.6)	(3.2)	
85	Alcohols (1-Hexanol, nonanol), ester frag.	10 ^a^	9.7 ^a^	16 ^b^	15 ^b^	0.00	1.4 ^a^	5.1 ^b^	5.6 ^b^	3.5 ^ab^	0.00
		(1.5)	(1.9)	(1.7)	(1.1)		(0.7)	(0.9)	(0.6)	(1.1)	
87	Frag. (pentanal, 2-pentanone)	1.1 ^ab^	2.4 ^c^	1.5 ^b^	0.8 ^a^	0.00	0.3 ^a^	0.8 ^b^	0.8 ^b^	0.5 ^a^	0.00
		(0.2)	(0.3)	(0.4)	(0.1)		(0.1)	(0.2)	(0.1)	(0.1)	
89	Butyrate related esters (ethyl butanoate, propyl butanoate, butyl butanoate)	26 ^a^	39 ^ab^	59 ^b^	16 ^a^	0.00	2.6	18	6.1	5.8	0.05
		(9.5)	(9)	(15)	(2.1)		(0.7)	(12)	(1.4)	(0.7)	
95	Farnesene frag.	0.9 ^a^	1.3 ^ab^	4.6 ^b^	0.3 ^a^	0.01	0.1	0.3	0.8	0.3	0.10
		(0.7)	(0.6)	(2)	(0.5)		(0.2)	(0.3)	(0.3)	(0.2)	
99	Aldehydes (trans-2-hexanal), esters (ethyl hexanoate, hexyl acetate)	5.4 ^a^	11 ^b^	11 ^b^	3.4 ^a^	0.00	3.3 ^a^	7.0 ^b^	4.1 ^ab^	2.3 ^a^	0.00
		(1.1)	(1.9)	(2.6)	(0.9)		(0.8)	(0.8)	(1.8)	(0.6)	
101	Aldehydes(2-hexanone, hexanal)	0.8	1	1.7	0.7	0.12	0.3 ^a^	0.9	0.6 ^ab^	0.5 ^ab^	0.01
		(1.5)	(0.4)	(0.7)	(0.2)		(0.1)	(0.2)	(0.2)	(0.1)	
103	Esters (isoamyl esters, propyl acetate, ethyl 2-methyl butanoate, methyl butanoate)	18 ^a^	29 ^b^	9.7 ^a^	20 ^ab^	0.00	3.2 ^a^	17 ^b^	6.6 ^a^	8.4 ^a^	0.00
		(1.1)	(6.1)	(2.1)	(4.3)		(1)	(3.8)	(3.6)	(2.9)	
109	Unidentified	1.4 ^a^	1.5 ^a^	4.1 ^b^	0.7 ^a^	0.00	0.7	0.6	0.8	0.4	0.34
		(0.3)	(0.5)	(1.1)	(0.2)		(0.1)	(0.2)	(0.4)	(0.1)	
117	Esters (hexanoates, ethyl 2-methyl butanoate, Isobutyl acetate, butyl acetate)	22 ^a^	25 ^a^	44 ^b^	18 ^a^	0.00	2.3 ^a^	9.2 ^b^	4.0 ^ab^	6.6 ^ab^	0.02
		(4.3)	(2.6)	(3.7)	(3.3)		(1.1)	(3.7)	(1.3)	(1.6)	
121	Acetophenone	0.7 ^a^	0.9 ^a^	3.7 ^b^	0.4 ^a^	0.00	0.1	0.5	0.4	0.3	0.35
		(0.3)	(1)	(1.2)	(0.2)		(0.1)	(0.5)	(0.2)	(0.1)	
123	Farnesene frag.	0.4 ^a^	1.1 ^a^	2.3 ^b^	0.3 ^a^	0.00	0.3	0.4	0.5	0.3	0.58
		(0.3)	(0.7)	(1.4)	(0.5)		(0.3)	(0.3)	(0.2)	(0.2)	
131	Esters (heptanoates, methyl hexyl-esters)	1.4 ^b^	3.1 ^c^	0.6 ^a^	1.2 ^ab^	0.00	0.4	0.9	0.5	0.6	0.35
		(0.4)	(0.1)	(0.2)	(0.3)		(0.2)	(0.6)	(0.2)	(0.1)	
135	farnesene frag., P-cymene	0.5 ^a^	0.7 ^a^	3.4 ^b^	0.2 ^a^	0.00	0.3	0.4	0.3	0.1	0.80
		(0.4)	(0.4)	(0.3)	(0.4)		(0.2)	(0.6)	(0.5)	(0.1)	
137	Monoterpenes, farnesene frag.	0.7 ^a^	0.3 ^a^	3.2 ^b^	0 ^a^	0.00	0.1	0	0.3	0.1	0.28
		(0.7)	(0.5)	(0.4)	(0.1)		(0.1)	(0.1)	(0.3)	(0.2)	
145	Esters (ethyl hexanoate, butyl butanoate)	2.4 ^a^	2.9 ^a^	4.3 ^b^	3.1 ^ab^	0.01	0.3	1.1	0.4	0.8	0.20
		(0.3)	(0.1)	(0.9)	(0.4)		(0.3)	(0.8)	(0.2	(0.2)	
149	Farnesene frag., Phenyls (estragole, anethol)	1.9 ^a^	0.6 ^a^	5.3 ^b^	4.0 ^a^	0.01	0.3	0.7	0.2	0.3	0.57
		(0.1)	(0.4)	(1.4)	(2.1)		(0.3)	(0.6)	(0.2)	(0.4)	
173	Decanoates	1.0 ^ab^	1.4 ^ab^	2.2 ^b^	0.8 ^a^	0.03	0.2	0.2	0.3	0.4	0.46
		(0.4)	(0.1)	(0.8)	(1.4)		(0.1)	(0.2)	(0.1)	(0.1)	
205	Alpha-farnesene	1.4 ^a^	1.8 ^a^	7.3 ^b^	0.4 ^a^	0.00	0	0.6	0.3	0.1	0.02
		(0.3)	(1.1)	(2.3)	(0.4)		(0)	(0.1)	(0.3)	(0.1)	

**Table 3 foods-12-04050-t003:** Significant change (*p* < 0.05) during storage for the concentration of selected esters (Figure 3), aldehydes (Figure 4) and alcohols (Figure 5) for either untreated or 1-MCP-treated (SF) apples are indicated by lowercase (untreated) or uppercase (treated) letters. Days that have different letters for each cultivar had a significant change during storage.

			Cultivar	Untreated (days)					Treated (SF) (days)				
				50	70	100	120	150	50	70	100	120	150
Ester	*m*/*z* 61	Acetate-related esters	Braeburn	A	ab	ab	B	b	A	A	A	A	B
			Fuji	A	ab	b	ab	ab	-	-	-	-	-
			Golden Delicious	Ab	ab	ab	bc	c	-	-	-	-	-
			Jazz	-	-	-	-	-	-	-	-	-	-
	*m*/*z* 89	Butyrate-related esters	Braeburn	-	-	-	-	-	A	A	A	A	B
			Fuji	a	ab	bc	bc	c	AB	A	AB	BC	C
			Golden Delicious	b	ab	ab	ab	ab	-	-	-	-	-
			Jazz	-	-	-	-	-	-	-	-	-	-
	*m*/*z* 117	Hexanoate-related esters	Braeburn	-	-	-	-	-	A	A	A	A	B
			Fuji	a	ab	ab	ab	b	A	AB	AB	AB	B
			Golden Delicious	bc	ab	b	bc	c	A	A	A	B	B
			Jazz	-	-	-	-	-	-	-	-	-	-
	*m*/*z* 145	Ethyl hexanoate/butyl butanoate	Braeburn	a	ab	ab	b	b	A	A	A	A	B
			Fuji	a	ab	ab	ab	b	-	-	-	-	-
			Golden Delicious	ab	a	ab	ab	b	A	A	A	B	B
			Jazz	a	ab	ab	b	b	-	-	-	-	-
Aldehyde	*m*/*z* 45	Acetaldehyde	Braeburn	a	a	b	b	c	A	A	A	B	C
			Fuji	-	-	-	-	-	A	B	AB	AB	B
			Golden Delicious	-	-	-	-	-	C	A	AB	B	BC
			Jazz	a	b	bc	c	d	A	AB	AB	B	AB
	*m*/*z* 73	Butanal	Braeburn	a	ab	b	b	b	A	A	A	A	B
			Fuji	-	-	-	-	-	A	A	A	AB	B
			Golden Delicious	-	-	-	-	-	A	B	A	A	AB
			Jazz	-	-	-	-	-	-	-	-	-	-
	*m*/*z* 101	Hexanal/2-hexanone	Braeburn	a	ab	b	b	b	A	A	A	A	B
			Fuji	a	ab	ab	b	b	-	-	-	-	-
			Golden Delicious	-	-	-	-	-	-	-	-	-	-
			Jazz	a	ab	ab	ab	b	A	A	AB	B	AB
Alcohol	*m*/*z* 33	Methanol	Braeburn	a	ab	ab	b	b	A	A	A	A	B
			Fuji	-	-	-	-	-	A	AB	AB	B	B
			Golden Delicious	-	-	-	-	-	A	A	B	B	B
			Jazz	a	ab	bc	bc	c	-	-	-	-	-
	*m*/*z* 47	Ethanol	Braeburn	a	a	b	b	b	A	A	A	A	B
			Fuji	a	a	ab	ab	b	-	-	-	-	-
			Golden Delicious	a	ab	ab	ab	b	C	A	B	B	C
			Jazz	a	ab	bc	c	d	A	AB	AB	B	AB

**Table 4 foods-12-04050-t004:** Pearson’s correlation coefficients calculated for physico-chemical, texture and µ-CT data. Highlighted boxes indicate a significant positive correlation (>0.6; α < 0.05) (green) and a negative correlation (<−0.6; α < 0.05) (red).

	DM	SSC	Fmax	Flesh.f	Gradient
DM	-				
SSC	0.789	-			
Fmax	0.244	0.132	-		
Flesh.f	0.116	0.029	0.900		
Gradient	−0.006	0.001	0.468	0.520	-
AUC.peel	0.316	0.297	0.189	0.014	−0.713
AUC.flesh	−0.123	−0.276	0.773	0.861	0.497
Porosity	0.005	0.187	−0.691	−0.715	−0.484
Anisotropy	−0.156	−0.382	0.435	0.468	0.284
Connectivity	−0.082	0.170	−0.319	−0.257	−0.066
	AUC.peel	AUC.flesh	Porosity	Anisotropy	
DM					
SSC					
Fmax					
Flesh.f					
Grad					
AUC.peel	-				
AUC.flesh	−0.188	-			
Porosity	0.183	−0.811	-		
Anisotropy	−0.191	0.654	−0.619	-	
Connectivity	−0.057	−0.356	0.493	−0.745	

## Data Availability

The data used to support the findings of this study can be made available by the corresponding author upon request.

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
