# Peer review of "The Physical and Structural Effects of 1-MCP on Four Different Apple Cultivars during Storage"

_foods, 2023, doi:10.3390/foods12224050_

Round 1

Reviewer 1 Report

Comments and Suggestions for Authors

The manuscript describes a complex research process of good quality. Design of the research activities and presentation of the results are appropriate. On the other hand, several issues shall be fixed. See my remarks below.

The title of manuscript seems to be somewhat confusing. From the content of the paper, it becomes clear that the effects of 1-MCP treatment (and cold storage) are studied. Please, reformulate it in order to have a clear and concise title.

All along the paper the space between the abbreviation m/z and the subsequent number is missing: "m/z117" Precisely this should look: m/z 117.  Please, modify everywhere!

Page 1, rows 45-47: “Previous research on the impact of apple microstructure has generally focused on how the morphological size and orientation of the intercellular spaces (IS) influenceS texture [14,15].” Please, delete the highlighted ‘S’.

Page 2, rows 57-60: “For stored apples, 1-MCP is speculated to delay the expansion of schizogenous IS it delays the solubilization of polyuronides and neutral sugars [18] and reduces the activity of enzymes (polygalacturonase, pectin methylesterase, cellulase, β-galactosidase, and α-L-arabinofuranosidase) that cause cell wall degradation [19].”

Something seem to be missing from this sentence since it is very hard to decipher in its present form. Please, check and modify.

Page 3, rows 109-110: “Texture measurements were carried out in triplicate and three apples from each cultivar was assessed.”

… from each cultivar WERE assessed.

Page 3, rows 118-119: “The concentration of dry matter (DM %) was calculated as the percentage of weight after drying divided by the weight prior to drying.”

This sentence is not understandable; please, modify it.

Page 4, rows 158-159: “Headspace analysis of the PTR-MS data was converted from cps to concentration (ppbv) in accordance to Lindinger and others [27].”

First, it may be assumed that not the PTR-MS data went under the headspace analysis, but the apple pieces theirselves. Second, cps is not a widely used abbreviation, ppbv is not a well interpretable concentration unit. Las, but not least, the majority of prospective readers may not be so familiar with the work of ‘Lindinger and others’. Please, modify the sentence accordingly, and add at least some details on the method utilised.

Page 4, rows 165-166: “and the ethylene concentration was estimated according to Cappellin and others [28].”

The same remark as above for Lindinger’s work. Please, describe briefly the method you used.

Pages 7-9, Table 2 caption: “Different superscripts and p-values for the each VOC indicate the significant differences between cultivars based on the untreated and treated cultivars respectively”

This sentence is very hard to decipher. When looking to the data in the table, we see the p values, as well as, ‘a’ and ‘b’ superscripts, and it is hard to find out their exact meaning. Please, formulate this on a more precise way.

Page 10, rows 228-230: “The evolution of important ester related VOCs (m/z61, 89, 117, 145) that impart fruity flavors during consumption [31,39] were studied over postharvest storage time (Figure 3) where significant changes at specific time points for each cultivar are shown in Table 3.”

This sentence is not understandable. Please, revise.

Page 12, Table 3 caption: The description of the letters in the table (i. e. a, bc, or AB) are not clear enough. Please, modify in order to enhance understandability.

Page 17, row 400: “This affect increased the probability of the occurrence of intercepts between IS and cell walls." Change the word “affect" to "effect".

Page 18, rows 432- 436: connectivity and porosity _were_. Please, correct.

Page 18, rows 467-470: "The Rv coefficient for the untreated samples between days 50 and 100 was 0.73. The coefficient increased for days 100 and 150 (Rv=0.89) indicating that, regardless of cultivar, VOC emission and texture was more different between days 50 and 100 for untreated apples than between 100 to 150 days."

It would be very interesting to see these coefficients also for the treated samples.

Page 20, row 504: "Porosity and connectivity was ... " _ were _

Page 20, row 512:  "Nevertheless, SF treated samples at day 150 was ... " _were_

Comments on the Quality of English Language

The general English of the paper is acceptable, minor corrections shall be made (see my remarks).

Author Response

Reviewer 1

Comments and Suggestions for Authors

The manuscript describes a complex research process of good quality. Design of the research activities and presentation of the results are appropriate. On the other hand, several issues shall be fixed. See my remarks below.

The title of manuscript seems to be somewhat confusing. From the content of the paper, it becomes clear that the effects of 1-MCP treatment (and cold storage) are studied. Please, reformulate it in order to have a clear and concise title.

The title has been changed and now reads

“The physical and structural effects of 1-MCP on four different apple cultivars during cold storage”.

All along the paper the space between the abbreviation m/z and the subsequent number is missing: "m/z117" Precisely this should look: m/z 117.  Please, modify everywhere!

This has now been modified throughout the manuscript.

Page 1, rows 42-44: “Previous research on the impact of apple microstructure has generally focused on how the morphological size and orientation of the intercellular spaces (IS) influenceS texture [14,15].” Please, delete the highlighted ‘S’.

Row 43: This has now been modified.

Page 2, rows 88-90: “For stored apples, 1-MCP is speculated to delay the expansion of schizogenous IS it delays the solubilization of polyuronides and neutral sugars [18] and reduces the activity of enzymes (polygalacturonase, pectin methylesterase, cellulase, β-galactosidase, and α-L-arabinofuranosidase) that cause cell wall degradation [19].”

Something seem to be missing from this sentence since it is very hard to decipher in its present form. Please, check and modify.  

We apologise “as” was missing between …. IS “as” it ……. We have however, rewritten this sentences as below

Rows 88-90 have now been modified to the following:

For stored apples, 1-MCP is speculated to delay the solubilization of polyuronides and neutral sugars [18] and reduce the activity of enzymes (polygalacturonase, pectin methylesterase, cellulase, β-galactosidase, and α-L-arabinofuranosidase) that cause cell wall degradation [19].

Page 3, rows 144-145 “Texture measurements were carried out in triplicate and three apples from each cultivar was assessed.”

… from each cultivar WERE assessed.

 Rows 144-145 have now been change to the following:

Texture measurements were carried out in triplicate and three apples from each cultivar were assessed.

Page 3, rows 153-154: “The concentration of dry matter (DM %) was calculated as the percentage of weight after drying divided by the weight prior to drying.”

This sentence is not understandable; please, modify it.

 Rows 153-154 have been changed to the following:

Dry matter concentration (DM %) was calculated by dividing the weight of the apple after drying by its weight before drying and expressing it as a percentage.

Page 4, rows 199-202: “Headspace analysis of the PTR-MS data was converted from cps to concentration (ppbv) in accordance to Lindinger and others [27].”

First, it may be assumed that not the PTR-MS data went under the headspace analysis, but the apple pieces themselves. Second, cps is not a widely used abbreviation, ppbv is not a well interpretable concentration unit. Las, but not least, the majority of prospective readers may not be so familiar with the work of ‘Lindinger and others’. Please, modify the sentence accordingly, and add at least some details on the method utilised.

Rows 199-202 have been modified to the following for clarity:

The PTR-MS data from the headspace analysis of the apple pieces was converted from counts per second (cps) to concentration in parts per billion per volume (ppbv). This was done in accordance to Lindinger and others [27] assuming the proton transfer reaction rate constant to be 2 × 10−9 cm3/s for all compounds. This allows an estimation of concentration without calibration with a possible systematic error of about 20%.

Page 4, rows 209-210: “and the ethylene concentration was estimated according to Cappellin and others [28].”

The same remark as above for Lindinger’s work. Please, describe briefly the method you used.

Rows 209-210have been modified with additional information for clarity. It now reads:

Due to ethylene’s low proton affinity, it is not protonated by H3O+ but rather ionised by charge transfer from O2+ (note O2+ is present in low levels in H3O+).  The concentration of ethylene present was therefore calculated according to Cappellin and others [28] by measuring the parent ion at m/z 28 and using a reference standard to calibrate and provide an accurate real-time estimate in ppbv.

Pages 7-9, Table 2 caption: “Different superscripts and p-values for the each VOC indicate the significant differences between cultivars based on the untreated and treated cultivars respectively”

This sentence is very hard to decipher. When looking to the data in the table, we see the p values, as well as, ‘a’ and ‘b’ superscripts, and it is hard to find out their exact meaning. Please, formulate this on a more precise way.

Table 2 caption has been modified to the following for clarity:

P-values for each VOC are listed comparing the untreated and treated cultivars. For significant p-values (p < 0.05), different superscripts indicate significant differences between cultivars.

Page 10, rows 285-287: “The evolution of important ester related VOCs (m/z61, 89, 117, 145) that impart fruity flavors during consumption [31,39] were studied over postharvest storage time (Figure 3) where significant changes at specific time points for each cultivar are shown in Table 3.”

This sentence is not understandable. Please, revise.

Rows 285-287 has been modified for clarity:

The evolution of the key fruity ester related compounds (m/z 61, 89, 117, 145) were monitored over postharvest storage time (Figure 3), with significant changes in VOC content over time for each cultivar being reported in Table 3.

Page 12, Table 3 caption: The description of the letters in the table (i. e. a, bc, or AB) are not clear enough. Please, modify in order to enhance understandability. 

Table 3 caption has been modified to the following for clarity:

Significant changes (p <0.05) during storage in the concentration of selected esters (Figure 3), aldehydes (Figure 4) and alcohols (Figure 5) for either untreated and treated (SF) apples are indicated by lowercase (untreated) or uppercase (treated) letters. Days that have different letters for each cultivar had a significant change during storage.

Page 17, row 4477: “This affect increased the probability of the occurrence of intercepts between IS and cell walls." Change the word “affect" to "effect".

Row 477 has been changed as suggested.

Page 18, rows 512: connectivity and porosity _were_. Please, correct.

Row 512 has been modified as suggested.

Page 18, rows 467-470: "The Rv coefficient for the untreated samples between days 50 and 100 was 0.73. The coefficient increased for days 100 and 150 (Rv=0.89) indicating that, regardless of cultivar, VOC emission and texture was more different between days 50 and 100 for untreated apples than between 100 to 150 days."

It would be very interesting to see these coefficients also for the treated samples.

The discussion of the Rv coefficients for the treated sample has been addressed in rows 560-566.   CHECK 580 – 590.  Note that in the original text we referred to trated apples as SF 50, SF100 or SF150.  We have now inserted the word “treated” to make it clearer to the reader the SF (Smartfresh) refers to “treated” apples/

In Figure 8, the treated Jazz and Braeburn cultivars were separated from Golden Delicious and Fuji on Dim 2. The small distance between TREATED Jazz and Braeburn suggests strong similarities in terms of VOC release, morphology, texture and physico-chemical properties compared to Golden Delicious and Fuji. The Jazz cultivar also showed little change in the aforementioned attributes as shown by the small distances between the colored dots that represent storage time. The Rv coefficient between SF50 and SF100 was 0.92 indicating that TREATED apples changed very little within that time frame. However, the data between SF100 and SF150 were less similar (Rv=0.71) suggesting changes occurred in VOC release and texture data during this time in at least some apple cultivars. When comparing both SF and untreated samples, both were similar on day 50 (Rv=0.81). Based on different time points, C100 was most similar to SF150 (Rv=0.95) suggesting that SF treatment could lengthen the shelf life of samples by at least 6 weeks.

Add in Text

Page 18, 20, row 512, 585: "Porosity and connectivity was ... " _ were _ 

The change has been made for the respective pages and rows for consistency throughout the manuscript.

Page 20, row 593:  "Nevertheless, SF treated samples at day 150 was ... " _were_ 

Corrected as suggested.

Reviewer 2 Report

Comments and Suggestions for Authors

Minor corrections

The abstract needs to be revised as there was no clear conclusion of the study. Figures 2-5 should also be corrected to reflect the control and treated samples clearly.

Line 12: The degree sign should be corrected and in subsequent appearance.

Line 15-17: “as observed in previous studies” This phrase should be removed, and the sentence should also be rephrased.

Line 25: “Keywords” is missing.

Author Response

Comments and Suggestions for Authors

Minor corrections

The abstract needs to be revised as there was no clear conclusion of the study.

The abstract has been revised for clarity to show clear findings at the end.

The impact of the ethylene inhibitor, 1-methylcyclopropene (1-MCP) on four apple cultivars (Braeburn, Fuji, Jazz and Golden Delicious) over 150 days of storage at 2 °C was assessed. Proton transfer reaction quadrupole mass spectrometry (PTR-QUAD-MS) was used to monitor changes in VOC composition, while texture analysis and X-ray microcomputer tomography (µ-CT) scanning were used to study microstructural changes.

In summary, the application of 1-MCP on apples reduced VOC emissions concurrently maintaining a firmer texture compared to the untreated apples at each time point. The µ-CT scanning revealed how changes in specific morphological characteristics such as anisotropy, connectivity and porosity, size and shape, as well as interconnectivity of intracellular spaces (IS) influenced texture even when porosity was similar. Additionally, this study showed that porosity and the connectivity of IS were associated with VOC emission and increased simultaneously. This study highlights how the morphological parameters of an apple can help explain their ripening process during long term storage and how their microstructure can influence the release of VOCs.

Figures 2-5 should also be corrected to reflect the control and treated samples clearly.

The authors feel the figures are already well labelled. In figure 2, it was indicated the untreated apples are shown as continuous lines whereas the treated samples, as dotted lines.

In Figures 3-5, clear indication of the untreated apples are stated on the left panel and the SF treated apples on the right panel is written in the caption. The consistency of using continuous lines for untreated apples and dotted lines for treated apples is shown for these figures.

Line 11: The degree sign should be corrected and in subsequent appearance.

This has been changed for lines 10, 101, 172, 338.

Line 14-16: “as observed in previous studies” This phrase should be removed, and the sentence should also be rephrased.

The abstract has been rewritten for clarity as suggested by Reviewer 2.

Line 24: “Keywords” is missing.

This has been added to line 22.

Reviewer 3 Report

Comments and Suggestions for Authors

This manuscript deals with very important topic. The entire experimental setup covers vast range of analyses, but systematically determined in order to deeply examine the influence of 1-MCP and cold storage on quality of apple fruits.

The results are clearly presented, and an ease in reading of this manuscript may draw huge readers' attention.  Just pay attention to some errors and typos throughout the manuscript (i.e. line 204: ...reported for apples [36,37] AND apple juice [8]; line 278: However, hexanal/2-hexanone...).  

Author Response

Comments and Suggestions for Authors

This manuscript deals with very important topic. The entire experimental setup covers vast range of analyses, but systematically determined in order to deeply examine the influence of 1-MCP and cold storage on quality of apple fruits.

The results are clearly presented, and an ease in reading of this manuscript may draw huge readers' attention.  Just pay attention to some errors and typos throughout the manuscript (i.e. line 256: ...reported for apples [36,37] AND apple juice [8]; line 349: However, hexanal/2-hexanone...).  

These minor corrections have been made

Reviewer 4 Report

Comments and Suggestions for Authors

This manuscript is well structured and well written, with clear presentation of the results and almost nothing to revise. But as authors suggested, there is a huge data of similar investigations and the paper does not offer novelty. It offers new technologies in the analysis of the apples and statistical correlations of the factors, so the dilemma is whether it should be published as original research paper or maybe as technical paper. The small remarks that I have are:

- why did you start at the day 50 and not sooner? if you had information from previous research that there are no changes till then you should state them or give proof.

- why did you not show the images from the CT since it is the technique that supports your results?

- in the caption of Figure 4 please check is it Table 3 or 4 correct at the end.

Author Response

Comments and Suggestions for Authors

This manuscript is well structured and well written, with clear presentation of the results and almost nothing to revise. But as authors suggested, there is a huge data of similar investigations and the paper does not offer novelty. It offers new technologies in the analysis of the apples and statistical correlations of the factors, so the dilemma is whether it should be published as original research paper or maybe as technical paper. The small remarks that I have are:

- why did you start at the day 50 and not sooner? if you had information from previous research that there are no changes till then you should state them or give proof.

The study started at day 50 to replicate the likely time when consumers would first be able to purchase apples that have been treated with 1-MCP.

This explanation has been added to the text on L 114

- why did you not show the images from the CT since it is the technique that supports your results

CT images of apples have been shown in an earlier paper ( Ting, V.J.L.; Silcock, P.; Bremer, P.J.; Biasioli, F.. X-ray micro-computer tomographic method to visualize the microstructure of different apple cultivars. J. Food Sci. 2013, 78(11):E1735-42.).

We have added a note to the  current paper L ???  stating that

Examples of images obtained using µ-CT scanning can be seen in Ting et al 2013 [17]. 

We do not propose to add images to the current paper as the MS already contains a large number of Figures and the images are not as useful as the data that can be derived from them

- in the caption of Figure 4 please check is it Table 3 or 4 correct at the end.

This has now been changed to Table 3.